# Proton-Conducting Biopolymer Electrolytes Based on Carboxymethyl Cellulose Doped with Ammonium Formate

**DOI:** 10.3390/polym14153019

**Published:** 2022-07-26

**Authors:** M. I. H. Sohaimy, M. I. N. Isa

**Affiliations:** 1Energy Materials Consortium (EMC), Advanced Materials Team, Ionic & Kinetic Materials Research (IKMaR) Laboratory, Faculty of Science & Technology, Universiti Sains Islam Malaysia, Nilai 71800, Malaysia; ibnuhyqal@gmail.com; 2Advanced Nano Materials (AnoMa), Advanced Materials Team, Ionic State Analysis (ISA) Laboratory, Faculty of Science & Marine Environment, Universiti Malaysia Terengganu, Kuala Nerus 21030, Malaysia

**Keywords:** biopolymer, cellulose, polymer electrolyte, ammonium formate, ionic conductivity

## Abstract

In this work, CMC-AFT biopolymer electrolytes system was developed using Carboxymethyl cellulose (CMC) doped with varied amount (10–50 wt.%) of ammonium formate (AFT) in order to study the effect of AFT on the biopolymer-salt system. The chemical structure of the biopolymer was studied using Fourier-Transform infrared (FTIR) and X-ray diffraction (XRD). The interaction between the COO^−^ of CMC and the weakly-bound H^+^ of NH_4_^+^ AFT occurred at 1573 cm^−1^ as seen in FTIR analysis and the amorphous phase was found to increase with the addition of AFT as seen from XRD pattern. Both FTIR and XRD testing indicates that the AFT had disrupted the CMC crystalline structure. The ionic conductivity of the CMC-AFT biopolymer electrolytes increases and achieved the highest value of 1.47 × 10^−4^ S·cm^−1^ with the addition of AFT. The impedance measurement showed that the capacitive and resistive behavior inside the biopolymer diminished when 50 wt.% of AFT was added. Dielectric analysis confirmed the increased number of charge carriers is due to the increase in AFT composition. Further dielectric analysis showed the occurrence of conductivity relaxation peak thus affirmed the charge carriers’ ability to travel further to a longer distances when AFT composition increases from 10 to 50 wt.%. The dielectric properties confirmed the non-Debye behavior of the CMC-AFT biopolymer electrolytes.

## 1. Introduction

Biopolymer materials are interesting since they share similar characteristics with synthetic polymers, albeit with better prospects in terms of cost and safety. Biopolymers are abundant, virtually unlimited, and environmentally friendly, which makes them exciting materials to be applied in technology. The biomedical field and food packaging are two areas that have pushed biopolymer applications [1,2]. Due to these advantages, the energy storage technology field also needs to shift its attention towards utilizing biopolymers. As seen with increased demand for electric vehicles, due to the worldwide effort to reduce the global carbon footprint thanks to the Paris Agreement of 2016 [3], energy storage advancement is of immediate importance for ensuring cheaper and safer energy storage to achieve the targets set by the agreement [3,4].

Biopolymers can help to realize the development of a solid-state battery by replacing the liquid electrolyte used in current conventional energy storage with the solid polymer electrolyte [5]. The electrolyte is the medium that allows ions to move through from anode to cathode to complete the circuit when connected to a load. The idea of using polymeric material as an electrolyte was first introduced in the 1970s when the solid polymer (polyethylene oxide) exhibited improved ionic conductivity when complexed with ionic salt [6]. Thus, a similar approach can be applied with biopolymer materials. Cellulose, chitin, starch, alginate, and carrageenan are a few examples of biopolymers that can be utilized in energy storage technology. Cellulose-based materials are very compelling biopolymers since they are easily found all over the world and said to be the number one in terms of materials availability [7]. Carboxymethyl cellulose (CMC) is an excellent material for biopolymer electrolyte application since it can easily dissolve in water due to its hydrophilic properties compared to pure cellulose (water insoluble). This hydrophilic property negates the need for using an organic solvent such as N-Methyl-2-pyrrolidone (NMP), thus reducing hazard and cost. To top it off, the presence of the carbonyl group (C = O) within CMC is beneficial in electrolyte application as it can act as a coordination site for ion hopping.

For any practical electrochemical application there are few conditions that a biopolymer electrolyte need to achieve, such as high electrochemical stability window, good thermal stability in terms of electrical and physical properties, and high ionic conductivity. Ionic conductivity is the most important parameter, since high ionic conductivity helps to minimize Ohmic losses [8]. Therefore, researchers have proposed multiple methods to convert these insulating biopolymer materials into semi-conducting materials. The incorporation of inorganic filler materials, co-polymerization, doping with ionic salt, and the addition of plasticizer are some of the techniques that can be employed [9]. Doping with ionic salt is the simplest method to improve ionic conductivity by providing the necessary conducting ions (Li^+^, H^+^, K^+^ etc.). On top of that, ionic salt also helps to disrupt the internal chemical structure, thus increasing the amorphous phase in the polymers which is beneficial for ionic conduction as proven by multiple reports previously [10,11,12,13,14]. 

The selection of ionic salts is important, where the criteria for selection are the type of conducting ions in the salts (sodium based, lithium based, etc.) and the ionic salts’ lattice energy [15]. The lattice energy indicates the ability of the ionic salts to dissociate into free ions, and thus low lattice energy is preferable. However, a report by Brza et al., (2020) showed that ammonium salts (NH_4_BF_4_) with a very low lattice energy (<600 kJ/mol) tend to associate back hence forming ion agglomeration [15] thus, led to low ionic conductivity value. From our calculations of lattice energy using the Kapustinskii equation [16], the lattice energy of ammonium formate (AFT) is moderate (737 kJ/mol) and suitable to function as the ionic salt for the proton-conducting biopolymer electrolyte. Additionally, previous reports also suggest the promising potential of AFT in polymer electrolytes where the ionic conductivity increased towards 10^−3^ S·cm^−1^ [17,18,19]. The ionic conductivity measured in the doping system may be attributed to both cations and anions. Thus, the heavier formate anion (HCOO^−^) of AFT is expected to contribute less towards total ionic conductivity compared to some other ammonium halide anions (F^−^, Cl^−^).

To date, no study on CMC biopolymer with ammonium formate as an electrolyte has been reported. In this paper, a biopolymer electrolyte system from CMC was doped with varies amount of AFT. The CMC-AFT biopolymer electrolytes were studied to determine the effect of AFT composition on ionic conductivity, and structural properties. Additionally, dielectric analysis was performed to confirm the ionic conduction behavior.

## 2. Materials and Methods

### 2.1. Biopolymer Electrolyte Formation

The CMC (DS = 0.7) was procured from Across Organics/Thermo Fischer (CAS 9004-32-4, Geel, Belgium) and AFT salt from Sigma-Aldrich (CAS:540-69-2, Sigma–Aldrich Co., St. Louis, MO, USA). Both materials were used without any purification process. To produce the CMC-AFT biopolymer electrolytes, 1 g of CMC was first dissolved in distilled water. Once the CMC was completely dissolved, varied amount (10–50 wt.%) of AFT was added into the CMC solution. The solution was left to stir for 24 h to ensure complete salt dissolution. Then, the CMC-AFT solution was cast into glass petri dishes for the drying process. The dried samples were then kept in desiccators to prevent contamination and moisture absorption before further characterization. Table 1 shows the CMC-AFT biopolymer electrolyte composition for each sample.

### 2.2. Biopolymer Electrolyte Characterization

The ionic conductivity of the CMC-AFT biopolymer was determined using HIOKI IM3570 impedance analyzer. The samples were cut into suitable sizes and sandwiched between two stainless steel electrodes for testing. The samples were subjected to sinusoidal potential in between 50 Hz and1 MHz. The impedance data was plotted and analyzed for ionic conductivity and dielectric responses. The ionic conductivity was calculated using the following equation:(1)$$\sigma =\frac{t}{R_{b}\times A}$$
where *t* is sample thickness in cm, *R_b_* is the bulk resistance obtained from the intercept at X-axis of impedance plot (Ω), and *A* is the contact area between the sample and the stainless-steel electrodes. The dielectric value (dielectric constant, *ε_r_* and dielectric loss, *ε_i_*) was calculated using Equations (2) and (3).
(2)$$\varepsilon_{r}\left(\omega \right)=\frac{Z_{i}}{\omega C_{o}\left(Z_{r}{}^{2}+Z_{i}{}^{2}\right)},\;$$
(3)$$\varepsilon_{i}\left(\omega \right)=\frac{Z_{r}}{\omega C_{o}\left(Z_{r}{}^{2}+Z_{i}{}^{2}\right)}$$

The *Z_i_* and *Z_r_* in the equation are the imaginary and real impedance and *ω* is the angular frequency. Here, the *C_o_* = *ε_o_A/t* where *ε_o_* is the permittivity of free space, *A* is the contact area, and *t* is the sample thickness.

XRD brand RIGAKU Miniflex II was used to analyze the structure of the CMC-AFT biopolymer electrolytes. The samples were tested at an angle of 2*θ* = 5°–80° using Cuk*α* radiation (0.154 nm) source.

The FTIR spectroscopy measurement was carried out using Agilent Cary 630 equipped with an attenuated total reflection (ATR) accessory with a germanium crystal in the range of 3800 to 700 cm^−1^ with spectral resolution of 4 cm^−1^. FTIR data were recorded in the transmittance mode.

## 3. Results

### 3.1. FTIR Analysis

The pure AFT salt spectrum is shown in Figure 1, and it reveals several characteristic peaks. The vibration peak at ~770 cm^−1^ is due to the O = C–O deformation [20], while the vibration peak at ~1342 cm^−1^ is due to the C–H bending vibrations of formate ions (HCOO^−^) [21]. The doublet peaks at ~1422 cm^−1^ and ~1454 cm^−1^ are owing to the N–H bending vibration of ammonium ions (NH_4_^+^) [22,23]. The carboxylic (COO) group of the formate ions can be seen from the vibrations peak at ~1560 cm^−1^. The broad peak seen centered at ~2793 cm^−1^ corresponds to C–H vibration [24], while the two obvious shoulder peaks at ~2973 cm^−1^ and at ~3183 cm^−1^ correspond to N–H bending and N–H stretching, respectively [25]. The FTIR spectrum of the CMC powder is also shown in Figure 1. The large vibration peak seen at ~1017 cm^−1^ is the C–O vibration from the CMC side chain (CH_2_OCH_2_). The small vibration peaks at ~1321 cm^−1^ and ~1412 cm^−1^ are due to C–H bending and O–H scissoring. The carboxylic group (COO) vibration of the CMC can be seen at ~1585 cm^−1^, while the weak vibration peak seen in between ~2880–2910 cm^−1^ is due to C–H stretching in the CMC backbone, and the broad peak at ~3250 cm^−1^ is due to the O–H stretching [26,27,28,29,30].

Figure 2 shows the FTIR spectrum for AFT00–AFT50 samples. The FTIR spectrum of the CMC-AFT biopolymer was divided into several wavenumber ranges, (a) 3600–2600 cm^−1^, (b) 1700–1300 cm^−1^, (c) 1100–950 cm^−1^, and (d) 800–700 cm^−1^, to highlight the FTIR changes. It is noted that CMC film (AFT00) bears similar FTIR pattern as in powder form. From the figure, the addition of AFT exhibits changes when compared to the CMC film (AFT00) spectrum. In Figure 2a, the O–H vibrations centered at ~3220 cm^−1^ became prominent when the biopolymer was added with AFT. The peak also shifted slightly towards lower wavenumber (3194 cm^−1^) as the AFT composition increases (AFT50). This is believed to be due to the interaction between the AFT and CMC [18]. Figure 2b shows the C=O stretching from the COO of CMC at ~1585 cm^−1^. When AFT was added, the ~1585 cm^−1^ peak started shifting to ~1578 cm^−1^ for AFT10 and AFT20 and further shifted to ~1573 cm^−1^ for samples AFT40 and AFT50. The shifting of the COO peak to lower wavenumbers is believed to be due to the coordination with the dissociated H^+^ ions of the AFT salt, and indicates that hydrogen bonding has occurred. The subsequent addition of AFT from 10% to 50% meant more conducting ions were supplied into the biopolymer and thus increased the interaction with the COO. Thus, it can be assumed that in this work, the H^+^ ions have interacted with the CMC polar group (COO) when AFT was added into the biopolymer. This corroborates the O–H peak shifting (~3198 cm^−1^) observed before. In Figure 2c, the peak at 1017 cm^−1^ shows no significant shifting, but a new peak formed at ~1049 cm^−1^ for AFT10 and AFT20, and the peak shifted to 1055 cm^−1^. This peak is believed to be due to C–O from the HCOO^−^ of the AFT salt. New peaks that emerged at ~765 cm^−1^ as seen in Figure 2d, and at ~1345 cm^−1^ in Figure 2c, are from the O = C–O deformation and the C–H of the formate ion (HCOO^−^), confirming the incorporation of AFT into the biopolymer. Based on all these observations, the increased interaction between the NH_4_^+^ ions and the carboxylic group of CMC disrupts the polymeric chain structure, and this should subsequently increase the amorphous phase in the biopolymer. The improved amorphous phase results in lower energy barriers between coordination sites thus improves the rate of conducting ions hopping from one site to another. In the next section, observed amorphous phase changes are correlated with FTIR results.

### 3.2. XRD Analysis

The XRD spectra of the CMC powder and the AFT salt is shown in Figure 3a. The AFT salt shows multiple sharp peaks, which shows the crystalline nature of the AFT salt. On the other hand, the CMC powder shows a broad peak centered at 20°, which is the typical amorphous pattern seen in CMC similarly to previous reports [13,25,26,31]. Figure 3b shows the diffraction spectra of the CMC-AFT biopolymer electrolytes. The spectrum for CMC film (CMC-AFT00) showed a broad peak at about 22°, and no crystalline peak from the AFT was observed when AFT was added into the biopolymer (AFT10–AFT50), which suggests the complete dissolution of the AFT salt. The broad spectrum centered around 22° was observed for all samples (AFT10–AFT50), which indicated that the original crystalline structure of CMC was preserved during CMC-AFT biopolymer preparation and formation. According to Fahad et al., (2021), the peak centered at ~20–22° is due to the cellulose backbone chains that had little or no substituted group. In this case, it’s the cellulose backbone without the carboxymethyl group (–CH_2_OCH_2_COO^−^) structure which caused higher crystallinity [32].

Slight changes to the broad pattern were observed when AFT was added into the CMC. The peak center (22°) shifted to a lower angle (~20°) and broadened as the AFT was increasingly added into the CMC biopolymer. The broadened spectrum meant that the amorphous phase inside CMC biopolymer had increased. A small shoulder hump at (13°) was also observed, and this hump shifted to a lower angle (~11°). This hump occur due to the disruption of intramolecular hydrogen bonding of substituted cellulose chain structure (–CH_2_OCH_2_COO^−^) with backbone chain [32,33]. This shoulder hump changes are attributable to the coordination of the AFT salt towards the carboxylic group (COO) of the CMC sidechain as shown in the FTIR analysis. These interactions are likely to disrupt the inter-chain and intra-chain bonding between the polymeric backbones resulting in the increase of free volume inside the biopolymer, and thus an increased amorphous phase [18]. The increase amorphous phase will result in a higher ionic conductivity. The increase in the amorphous phase provides a higher number of vacant oxygens in the COO group for ionic conduction (H^+^) [34]. This will lead to improved ionic conduction hopping and subsequently improve the ionic conductivity of the CMC-AFT biopolymer electrolytes. It is observed that the ionic conductivity improves as the polymeric electrolytes amorphousness s increases, similarly demonstrated by previous studies [12,14,25,31].

### 3.3. Impedance Analysis

The impedance measurement (Cole–Cole plot) is presented in Figure 4. Typically, the impedance data consist of a straight line (low frequency) at an angle <90° to y-axis and semicircle at high frequency. In Figure 4, samples AFT00 up to AFT30 consist of both patterns while samples AFT40 and AFT50 only consist of a straight-line. This pattern can help to improve understanding of what happened inside the sample. To understand this, the impedance pattern can be represented with different electronic components such as a resistor (R) and a capacitor (C). The semi-circle (arc) observed from the plot represents the combination of resistive and capacitive components of the biopolymer electrolyte connected in parallel in an electric circuit, while the slanted straight line represents the constant phase element (CPE) of the biopolymer electrolyte connected in a series with a previous component [35]. All these electronic components can help to explain the ionic kinetics inside the biopolymer electrolytes. The capacitive behavior of the biopolymer electrolytes is expected due to the polymeric chain, which has been polarized by the electric field [36]. Since the carboxyl group is a polar functional group, the bulk capacitive is believed to be due to this group. On the other hand, the resistive behavior of the biopolymer electrolytes is due to the charge transfer process within the polymer bulk [36].

From the figure, semi-circle appearance was the largest for AFT00 and the size diminished above 30 wt.% of AFT. The shrinking of the semi-circle depicts less impedance faced by ionic species during the conduction process when AFT salt increases. On the other hand, the impedance plot for AFT40 and AFT50 shows that the semi-circle has completely disappeared with only the slanted/tilted line remaining. The disappearance of the semi-circle shows the effect of capacitive component inside the biopolymer electrolyte diminished, which in this case due to the polymeric chain polarization effect. This is in line with the FTIR results obtained previously where the interaction occurred at the carboxylic group of CMC, thus limiting the polarization effect. This also coincides with the XRD results, which shows the sample with the highest AFT composition (AFT50) has the highest amorphous phase. The ionic conductivity (Equation (1)) of the CMC-AFT biopolymer electrolytes can be calculated by extracting the bulk resistance (*R_b_*) value from the impedance plot. The *R_b_* value was obtained from the intercept between the high frequency semi-circle impedance and the slanted straight line. For AFT40 and AFT 50, which have no semi-circle pattern observed, the *R_b_* value was determined from the x-axis intercept by extending the straight line. It is noted that the *R_b_* value decreases, and the lowest value obtained is for sample AFT50.

### 3.4. Ionic Conductivity Analysis

Figure 5 shows the ionic conductivity trend at room temperature for the CMC-AFT biopolymer electrolytes. Ionic conductivity is the most important parameter where high ionic conductivity helps to minimize Ohmic losses [8]. The ionic conductivity behavior shows an increasing trend when the AFT salt composition is increased. The ionic conductivity for the AFT00 (undoped sample) biopolymer electrolyte calculated was 8.75 × 10^−9^ S·cm^−1^ and increased when added with 10% of AFT (AFT10). The ionic conductivity value continued to increase with increasing AFT composition, where the optimum value obtained for the CMC-AFT biopolymer electrolyte was at 1.47 × 10^−4^ S·cm^−1^ for AFT50. The increased value of ionic conductivity is due to the increase of charge carriers inside the CMC-AFT biopolymer electrolytes with increasing salt composition. The AFT salt dissociates into NH_4_^+^ and HCOO^−,^ and later interacts with the polar group of the CMC biopolymer. Therefore, higher degree of ion dissociation promotes higher ionic conductivity [34]. On top of that, the increased amorphous phase with the addition of AFT salt, as shown from previous XRD analysis, also helps to improve the ionic conductivity of the biopolymer electrolyte by lowering the energy barrier for ions to hop to the coordination site (COO) during the conduction process [31]. The highest conducting sample of the CMC-AFT biopolymer electrolyte (AFT50) exceeds the conventional molar ratio of the polymer electrolyte which is usually with a lower salt ratio. Nevertheless, a similar behavior was also observed previously in other reports [37,38]. The work done by Wei and Shriver (1998) also found similar behavior for their highest conducting sample (~10^−4^ S·cm^−1^) in their electrolytes system of (poly((1,3-dioxolan-2-one-4,5-diyloxalate) (PVICOX)) when doped with LiCF_3_SO_3_ at molar ratio of 1:1.

Srivastava et al. (1995) have reported previously that the cation transport in the polymer–ammonium salt system is only from H^+^ and not from other species based on Coulometric testing of their polymer–ammonium salt polymer electrolyte [39]. In this CMC-AFT case, the conducting ions came from the protonated (H^+^) of the ammonium ions NH_4_^+^. This process was most likely to occur due to the ability of one of the hydrogen atoms to protonate when an electric field was subjected across the biopolymer. The four hydrogen atoms in the tetrahedral structure of NH_4_^+^ were bonded differently, where one of the hydrogen atoms was tightly bonded, two hydrogen atoms were moderately bonded, and another one was loosely bonded to the N^+^ ion [31]. The loosely bonded hydrogen atom would protonate (H^+^) and hop from one coordination site to another site. With more AFT salt added into the polymer structure, more H^+^ was supplied into the polymer structure increasing the ionic conductivity of the CMC-AFT biopolymer electrolytes. The accepted consensus of the ionic conductivity value for any practical electrochemical application should be at least ≥10^−4^ S·cm^−1^ at room temperature [40], and the ionic conductivity measured in this work fulfilled that criterion. Table 2 shows the ionic conductivity of this work compared to other reports which use biopolymers in their electrolyte system. The optimum ionic conductivity of the CMC-AFT biopolymer electrolytes was found to be better than other biopolymer electrolytes reported previously, where most doped biopolymer electrolytes reported is in the range of ~10^−5^ S·cm^−1^.

### 3.5. Dielectric Analysis

Dielectric analysis can help to give further understanding of the ionic and molecular dynamics inside the biopolymer electrolytes, since most materials experience different types of polarization once induced by the electric field [31,46]. The dielectric properties of different biopolymer electrolytes were also affected by various internal and external factors such as an applied electric field, temperature, and molecular structure which is why dielectric analysis was done in this research study. Figure 6a shows the dielectric constant, *ε_r_* value at selected frequencies of the CMC-AFT biopolymer electrolyte, while Figure 6b shows the *ε_r_* value at various electric field frequencies. The *ε_r_* refers to the amount of trapped mobile ions at the electrode–electrolyte interface which formed a hetero-charge layer when induced by an electric field.

As seen in Figure 6a, the *ε_r_* value for the undoped sample (AFT00) is the lowest (0.07). This is due to most polarization arising on the side chain of the CMC (carboxylic group, COO). The dielectric constant, *ε_r_* value increases with addition of AFT and reaches a maximum value of 387 for AFT50. This correlated to the increased ionic dopant (AFT) added. Since the ammonium salts are polar molecules, this makes them susceptible to be polarized (dipole moment) when induced with an electric field. In this study, the AFT salt dissociated into individual cations (NH_4_^+^) and anions (HCOO^−^), then the weakly-bound H^+^ of NH_4_^+^ diffused across the biopolymer structure hence increase the *ε_r_* value. The increased dielectric value trend was also observed at higher frequencies (10 KHz and 100 KHz). Similar behaviors were also reported previously [47]. The dielectric value variation with changes in electric field frequency is shown in Figure 6b. From the figure, the *ε_r_* value were at the highest in the low frequency region. This is due to the ions’ ability to diffuse farther along the electric field to build up charges. On the other hand, the value decreases as the electric field frequency increases. This is due to the inability of the ions and molecules to rotate and align themselves to follow rapid changes of the electric field. This confined them to their localized position causing the cations and anions to interact and form neutral pairs, thus lowering the dielectric value. The dielectric value trend in this study with frequency confirms the non-Debye dependence of the CMC-AFT biopolymer electrolytes.

The investigation of dielectric modulus (*M*) analysis by using Equation (4) can also help elucidate other factors present in the CMC-AFT biopolymer electrolytes which affect its electrical properties. The advantage of modulus analysis is that it can suppress the charge accumulation at the blocking electrodes–electrolytes contact at low frequencies. According to Coşkun et al. (2018), dielectric modulus corresponds to the relaxation process occurring in the material physically [48]. In this research, the presence of polarizable polymer and ionic salt, polymer segmental relaxation, or ionic conductivity can play a role in ionic transport [49]. Figure 6c shows the real modulus (*M_r_*) plot for each of the CMC-AFT biopolymer samples. As seen from the figure, an obvious peak was observed across the frequency tested. This peak corresponds to the relaxation process inside the CMC-AFT biopolymer. The relaxation peaks reveal the ability of the charge carriers to hop from close-range to long-range. As clearly seen, the CMC-AFT biopolymer electrolytes’ relaxation peak appears to shift from lower frequency to higher frequency when AFT composition increases from AFT10 up to AFT30. This peak shift illustrates the increase number of free charge taking place during the conduction process and accumulated at the electrode–electrolyte interface as AFT composition increases [48]. The region at low frequency up to the maximum modulus peak value (relaxation peak) signifies the charge carriers’ ability to move freely over long distances. Meanwhile, at the higher frequency region, after the relaxation peaks, indicates the charge carriers are being confined to their potential wells resulting in their inability to move freely. This marks the point where the charge carriers transition from long-range to short-range with increased frequency [50,51]. The peak shifting also means different relaxation times constant for each sample. No relaxation peak was observed for samples AFT40 and AFT50, which was due to the limitation of frequencies tested. The low modulus value at low frequencies resulting in the appearance of a long tail for AFT40 and AFT50 indicates that the charge carriers’ ability to hop longer-range compared to other samples during the conduction process.
(4)$$M_{r}=\frac{\varepsilon_{r}}{\left(\varepsilon_{r}^{2}+\varepsilon_{i}^{2}\right)}$$

## 4. Conclusions

In this work, a CMC-AFT biopolymer electrolytes system was successfully developed via solution-casting technique. The ionic conductivity increases from 8.75 × 10^−9^ S·cm^−1^ and reaches optimum ionic conductivity of 1.47 × 10^−4^ S·cm^−1^. The ionic conductivity achieved in this work shows an encouraging prospect of the CMC-AFT biopolymer system to be used in electrochemical application. Through FTIR analysis, AFT salt appears to interact with the CMC biopolymer when added, as seen from the carboxylic group peak shifting to lower wavenumber (from 1585 cm^−1^ to 1573 cm^−1^). The addition of AFT salt into the CMC biopolymer also appears to disrupt the polymer backbone chain and increase the free volume inside the biopolymer electrolyte, thus increasing the amorphous phase of the biopolymer, as seen in the peak shifting and broadened XRD spectra. The increased ionic conductivity was found to be associated with increased charge carriers especially at high salt composition (AFT40 & AFT50). As more AFT salt added into the biopolymer, more ions available to be dissociated and participate in the ionic conduction, thus increasing the dielectric value. The effect of electric field frequencies reveals the ability of the charge carriers to hop longer-range when the AFT composition increases. Both dielectric and XRD analysis showed that the amorphousness of the biopolymer electrolytes system and the number of ionic charge supply contribute to the ionic conductivity improvement. Based on these results and observations, the ionic conduction process in this biopolymer electrolytes system can be explained–as the charge carriers (H^+^) dissociated from ammonium ions and hop from one coordinating site (C = O of carboxylic group of CMC) to another sites through the improved free volume inside the biopolymer. While the ionic conductivity obtained in this work is reasonable, it however sits on the borderline of possible electrochemical application (~10^−4^ S·cm^−1^). The CMC-AFT biopolymer can further be improved in future research by applying other techniques such as adding plasticizer or oxide filler materials. 

## Figures and Tables

**Figure 1 polymers-14-03019-f001:**
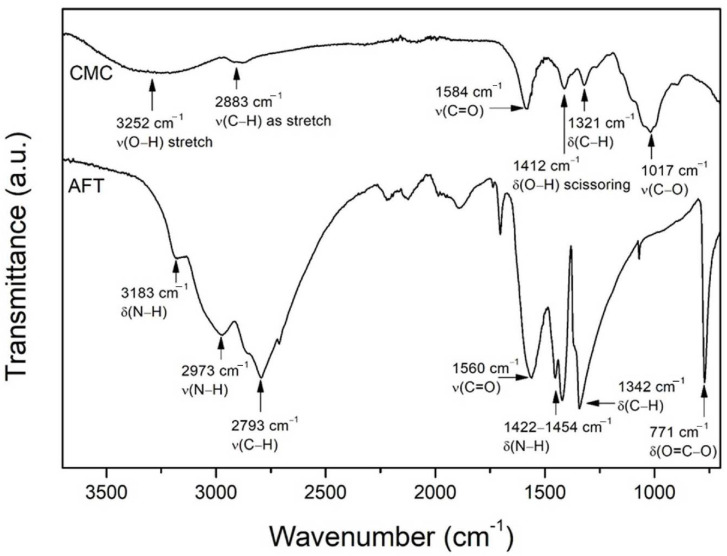
FTIR spectrum of CMC and AFT.

**Figure 2 polymers-14-03019-f002:**
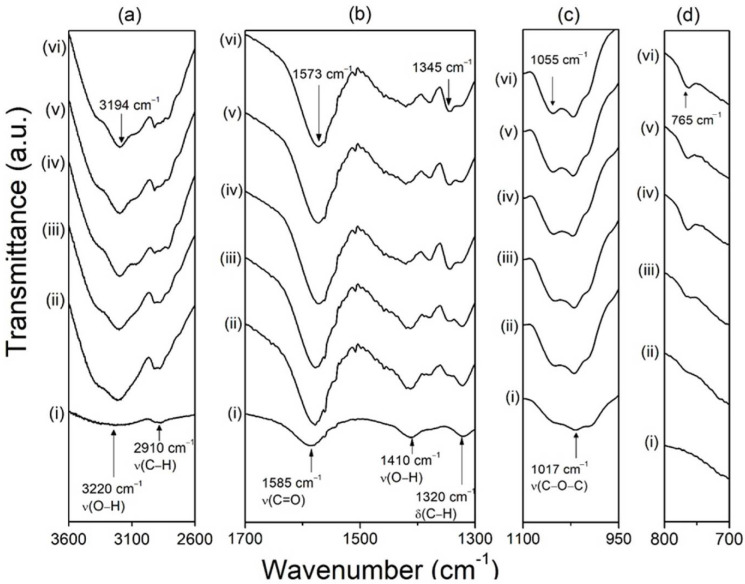
FTIR spectrum of CMC-AFT biopolymer electrolyte samples at different wavenumber range where (i) AFT00, (ii) AFT10, (iii) AFT20, (iv) AFT30, (v) AFT40 (vi) AFT50.

**Figure 3 polymers-14-03019-f003:**
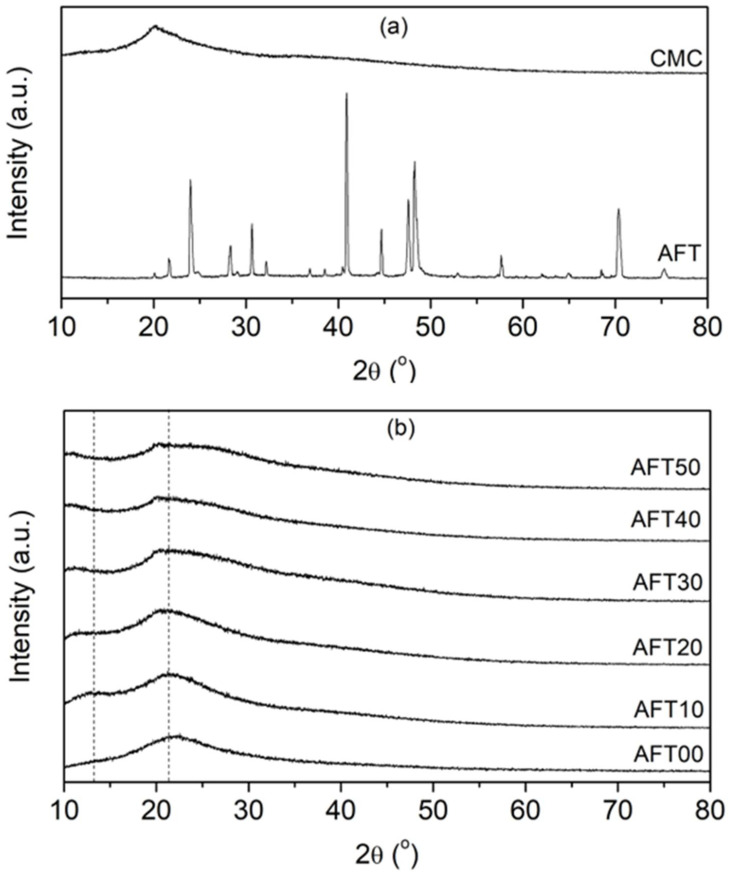
XRD pattern for (**a**) CMC powder and AFT used and (**b**) the CMC-AFT biopolymer electrolytes at different AFT composition.

**Figure 4 polymers-14-03019-f004:**
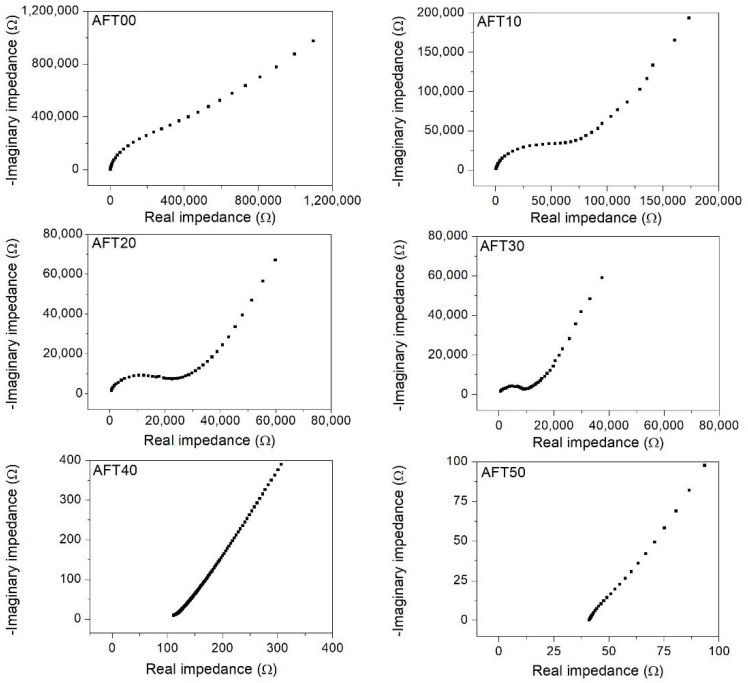
The impedance plot of the CMC-AFT biopolymer electrolytes.

**Figure 5 polymers-14-03019-f005:**
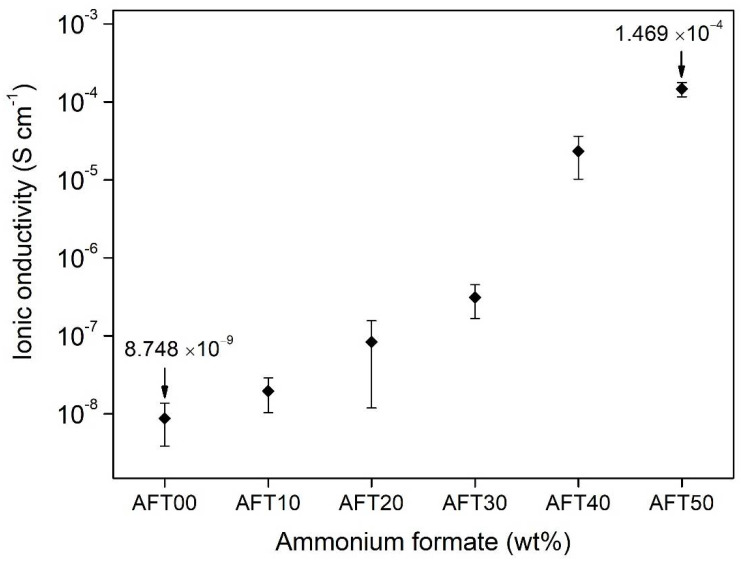
Room temperature ionic conductivity trends CMC-AFT biopolymer electrolyte at different AFT composition.

**Figure 6 polymers-14-03019-f006:**
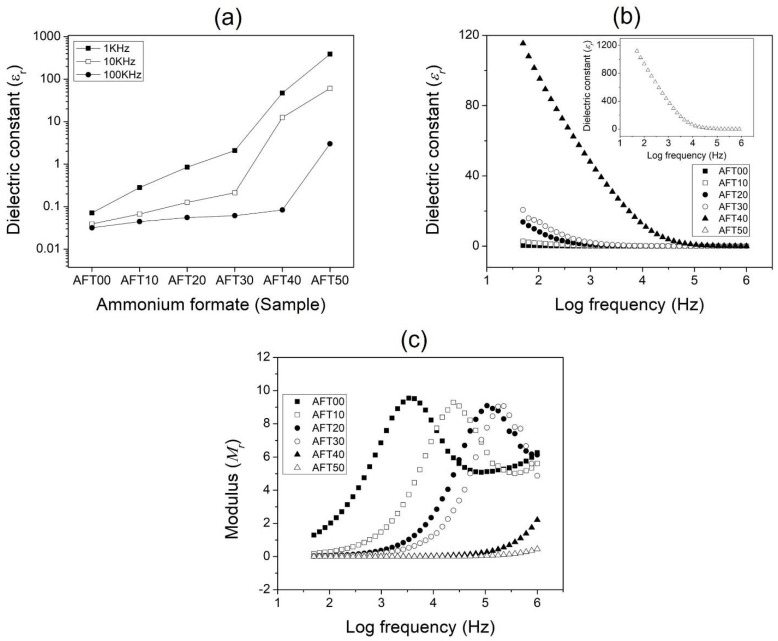
Dielectric constant plot of CMC-AFT biopolymer electrolytes at (**a**) selected frequency, (**b**) sweeping frequency and (**c**) dielectric modulus.

**Table 1 polymers-14-03019-t001:** CMC-AFT biopolymer materials composition and sample designation.

Designation	CMC (g)	AFT (g)	AFT (wt.%)
AFT00	1.000	-	0
AFT10		0.111	10
AFT20		0.250	20
AFT30		0.429	30
AFT40		0.667	40
AFT50		1.000	50

**Table 2 polymers-14-03019-t002:** Ionic conductivity of CMC-AFT biopolymer electrolyte with other biopolymers electrolyte.

Biopolymer Electrolyte	Ionic Conductivity, (S/cm)	References
CMC–DTAB	2.8 × 10^−5^	[41]
CMC–PVA-NH_4_Cl	8.86 × 10^−5^	[42]
Alginate-NH_4_Br	4.41 × 10^−5^	[43]
Chitosan-NH_4_I	1.11 × 10^−4^	[44]
Pectin-NH_4_SCN	4.05 × 10^−6^	[45]
CMC–AFT	1.47 × 10^−4^	Current work

## Data Availability

Not applicable.

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
