# Peer review of "Proton-Conducting Biopolymer Electrolytes Based on Carboxymethyl Cellulose Doped with Ammonium Formate"

_polymers, 2022, doi:10.3390/polym14153019_

Round 1

Reviewer 1 Report

The manuscript describes a novel conducting biopolymer electrolyte which
consists of carboxymethyl cellulose doped with ammonium formate. This
type of biopolymer electrolyte exhibits better electrical conductivity than
other biopolymer electrolytes and has a potential for use in batteries.

The paper is well written and provides sufficient details of both sample
preparation and measurements. The CMC-AFT films were studied with five
experimental techniques which complement each other.

Comments:

1. Table 1: AFT weight of 0.111 g for the AFT50 sample is likely wrong.

2. Figure 1: what does the "AFT" labelled curve (top) represent?

3. Figure 3: legend is missing on three out of six panels.

4. Are there any theoretical approaches available to describe biopolymer
electrolytes and some of their properties?   

Author Response

1. Table 1: AFT weight of 0.111 g for the AFT50 sample is likely wrong.

The value in Table 1 has been corrected

2. Figure 1: what does the "AFT" labelled curve (top) represent?

The AFT curve is for the pure AFT salt spectra. The AFT spectrum was presented in different figure from the rest of CMC-AFT biopolymer for better clarity.

3. Figure 3: legend is missing on three out of six panels.

The missing legend has been inserted in the figure.

4. Are there any theoretical approaches available to describe biopolymer
electrolytes and some of their properties?   

Theoretical approach related to the study is the lattice energy of doping salt and the amorphous phase of the biopolymer which was mentioned in the last paragraph of the introduction section and at XRD analysis section respectively.

Reviewer 2 Report

The idea of increasing of ionic-conductivity of biopolymers by using ionic dopants and plasticizers is well known and already published by the authors in https://doi.org/10.1016/B978-0-12-818134-8.00019-5 for example. That is why the new manuscript should contain either new insight on the biopolymer-based batteries or the thorough eperimental data which could be useful for the scientists. So this submission requires major revision.

1. Manuscript suffers of bad english so It is difficult to follow the data.

2. Abstract and Introduction should be rewritten completely. 

3. C=O is not carboxyl group but carbonyl.

4. Introduction. The first paragraph should be eliminated as different polymers have different physico-chemical properties, so the fragment "as flexble, good mechanical ability, good thermal stability, and most importantly, it is inexpensive " can not be applied to all of them.

5. "However, CMC biopolymers on its own are not capable to be applied in any electro- 59 chemical application since it has very low ionic conductivity (<10-7 S cm-1 )"  reference is needed

6. The choice of ammonium formate is not clear. The additional data on other dopants with indication on the reason of utilization of AFT is needed.

7. Usually CMC is supplied by manufacturer as sodium salt. The presence/ absence of Na+ ions should be clarified.

It is written that CMC was used without purification. That means that all impurities in the sample will affect the results. So, the control experiment with purified sample of CMC should be provided.

If this work will not be done the manuscript should be rejected as the data presented can not be named relible.

8. Figure 1. The IR-spectra are described poorly. No Analysis of COO- and COOH groups in CMC and composites is provided. The detailed analysis and impact of the charged groups on the conductivity should be presented. Please use the following reference to analyze the spectra - E. Pretsch, P. Bühlmann and C. Affolter, Structure Determination of Organic Compounds: Tables of Spectral Data, Springer, Berlin, Heidelberg, 2000 or any suitable reference.

9. Figure 3. Some experimental points do not fit the curves. So, the error bars, or indication of the error of the measurement, or splining of the curves should be presented.

Author Response

1. Manuscript suffers of bad english so It is difficult to follow the data.

The whole manuscript was revised for grammatical and language error.

2. Abstract and Introduction should be rewritten completely. 

The abstract and introduction has been edited as suggested.

3. C=O is not carboxyl group but carbonyl.

The grouping name was corrected as pointed out.

4. Introduction. The first paragraph should be eliminated as different polymers have different physico-chemical properties, so the fragment "as flexble, good mechanical ability, good thermal stability, and most importantly, it is inexpensive " can not be applied to all of them.

The paragraph was removed as suggested.

5. "However, CMC biopolymers on its own are not capable to be applied in any electro- 59 chemical application since it has very low ionic conductivity (<10-7 S cm-1 )"  reference is needed

The sentences has been revised as suggested by other reviewer.

6. The choice of ammonium formate is not clear. The additional data on other dopants with indication on the reason of utilization of AFT is needed.

Additional info was added in last paragraph of introduction section.

7. Usually CMC is supplied by manufacturer as sodium salt. The presence/ absence of Na+ ions should be clarified.

It is written that CMC was used without purification. That means that all impurities in the sample will affect the results. So, the control experiment with purified sample of CMC should be provided.

If this work will not be done the manuscript should be rejected as the data presented can not be named relible.

Sample designated as AFT00 is the control sample as the sample was CMC film without AFT salt. With regards to the Na+ ions, the contribution of Na in conductivity is negligible as seen from the value obtain and presented for AFT00. The degree substitution (DS) for CMC used in this research is 0.7 further shows insignificant contribution of Na as it is supposed to attached to carboxylic group. The XRD and FTIR results shows no significant deviation from other reports which shows no contamination of other impurities.  

8. Figure 1. The IR-spectra are described poorly. No Analysis of COO- and COOH groups in CMC and composites is provided. The detailed analysis and impact of the charged groups on the conductivity should be presented. Please use the following reference to analyze the spectra - E. Pretsch, P. Bühlmann and C. Affolter, Structure Determination of Organic Compounds: Tables of Spectral Data, Springer, Berlin, Heidelberg, 2000 or any suitable reference.

The Figure has been revised to better represents the data and rename as Figure 2. New figure (Figure 1) was added to show raw material IR spectra. Analysis of COO group of CMC was explained in second paragraph of the FTIR section.

9. Figure 3. Some experimental points do not fit the curves. So, the error bars, or indication of the error of the measurement, or splining of the curves should be presented

The sample AFT10 was retested and presented in Figure. No deviation was observed in new testing. The new testing value was used for dielectric analysis and the dielectric plot was revised as well using the new data.

Reviewer 3 Report

The article titled “Proton conducting biopolymer electrolytes based on carboxymethyl cellulose doped with ammonium formate” by Sohaimy et al. has been reviewed. The article is interesting to read and informative. However, the article does need some improvements. My specific comments are listed below.

11.       The English language needs correction in many places throughout the manuscript. Some of the sentences need to be rephrased while some needs grammatical corrections.

22.       Line No. 36. This sentence needs to be scientifically correctly presented. Biopolymer have already received huge attention in the scientific community not recently but being studied from decades. So, I could not agree with the word use “recently” in this sentence.

33.       English correction needed: Line Nos. 39-40; 42-43; 56-58; 68-69; 93-94; and many more sentences throughout the manuscript.

44.       Provide reference to the claim in Line Nos. 41-42.

55.       Line No. 54: “…..it can easily dissolve…..”. Provide the information of solvent here, “it can easily dissolve in what solvent?”

66.       Line no. 60: A specific data is not suitable here as the ionic conductivity of CMC changes with the concentration of the polymer, molecular weight, its degree of substitution, solvent, temperature, etc. cf. https://www.sciencedirect.com/science/article/abs/pii/S0021961416301082 Hence, either the detailed information has to be provided or the specific data can be removed.

77.       In Figure 1, it would be informative to insert the functional groups to some to the main peaks, at least, to present spectra in more informative way.

88.       It seems the naming of the materials are missing in the last three panels of Figure 3.

Overall the work is informative and interesting, however the above corrections are required.

Author Response

1. The English language needs correction in many places throughout the manuscript. Some of the sentences need to be rephrased while some needs grammatical corrections.

The manuscript has been revised accordingly.

2. Line No. 36. This sentence needs to be scientifically correctly presented. Biopolymer have already received huge attention in the scientific community not recently but being studied from decades. So, I could not agree with the word use “recently” in this sentence.

We agreed to your opinion about biopolymers. The sentences have been rephrased accordingly. Thank you for pointing that out.

3. English correction needed: Line Nos. 39-40; 42-43; 56-58; 68-69; 93-94; and many more sentences throughout the manuscript.

The lines refereed to and several other line has been revised.

4. Provide reference to the claim in Line Nos. 41-42.

The references have been added – reference number [3].

5. Line No. 54: “…..it can easily dissolve…..”. Provide the information of solvent here, “it can easily dissolve in what solvent?”

The sentence meant to say easily dissolve in water. The sentence has been revised accordingly.

6. Line no. 60: A specific data is not suitable here as the ionic conductivity of CMC changes with the concentration of the polymer, molecular weight, its degree of substitution, solvent, temperature, etc. cf. https://www.sciencedirect.com/science/article/abs/pii/S0021961416301082 Hence, either the detailed information has to be provided or the specific data can be removed.

The line has been revised (removed) as suggested.

7. In Figure 1, it would be informative to insert the functional groups to some to the main peaks, at least, to present spectra in more informative way.

The figure has been updated to show the functional group as suggested.

8. It seems the naming of the materials are missing in the last three panels of Figure 3.

The missing name has been inserted inside the figure.

Round 2

Reviewer 2 Report

Authors have improved the manuscript. However some important questions should be answered in revision.

1. The necessity of choice of ammonia format should be explained. 

in paper Ionic conductivity and electrical properties of carboxymethyl cellulose - NH4Cl solid polymer electrolytes. Journal of Engineering Science and TechnologyVol. 11, No. 6 (2016) 839 - 847 authors have demonstrated effective rise of the ionic conductance with lower share of the ammonia salt.

2. 50 wt.% of ammonia format significant exceeds the molar equivalency NH4+ to COO-. So the sharp rise of the conductivity at high ratio of the salt could be connected with increase of the share of free ions. Higher rate of the ammonia could increase conductivity more in this case. But could we speak about concductance of CMC in this case?

3. The sentence in Anstract "The interaction between AFT and CMC occurred at 1585 cm-1 the carbonyl (C=O) group of CMC with ammonium ions (NH4 + 17 ) of AFT" is incorrect. 

4. Authors neglect the 10-15 degrees regions in XRD spectra. It is obvious that salt-free sample differs from other. As well the XRD spectrum for ammonia format is not presented.

Author Response

1. The necessity of choice of ammonia format should be explained. 

Justification on selection of materials has been added in the text as suggested  by reviewer.

2. 50 wt.% of ammonia format significant exceeds the molar equivalency NH4+ to COO-. So the sharp rise of the conductivity at high ratio of the salt could be connected with increase of the share of free ions. Higher rate of the ammonia could increase conductivity more in this case. But could we speak about conductance of CMC in this case?

This system indeed exceeds the conventional polymer electrolyte molar ratio. However, similar behavior is observed in other reports and has been added in the text as reference – see ionic conductivity section. From dielectric analysis shows that increased conductivity is corelated towards high salt content while the polymer segmental effect on ionic conductivity is becoming less dominant as higher salt added as seen from dielectric modulus relaxation peak shifting.

3. The sentence in Abstract "The interaction between AFT and CMC occurred at 1585 cm-1 the carbonyl (C=O) group of CMC with ammonium ions (NH4 + 17 ) of AFT" is incorrect. 

The sentence has been rephrased.

4. Authors neglect the 10-15 degrees regions in XRD spectra. It is obvious that salt-free sample differs from other. As well the XRD spectrum for ammonia format is not presented.

XRD spectra for CMC powder and AFT crystal has been added. Explanation on the shoulder hump region (10-15 degree) was added in XRD section.

Thank you for the comments and suggestions given.

Reviewer 3 Report

The manuscript has been improved after the revisions made by the reviewers. Thus it may be accepted for publication.

Author Response

Thank you for the comments.

Round 3

Reviewer 2 Report

Authors have improved the manuscript and it could be published in present form.